# Implications of Glycosaminoglycans on Viral Zoonotic Diseases

**DOI:** 10.3390/diseases9040085

**Published:** 2021-11-17

**Authors:** Sarah Bauer, Fuming Zhang, Robert J. Linhardt

**Affiliations:** 1Department of Chemistry and Chemical Biology, Center for Biotechnology and Interdisciplinary Studies, Rensselaer Polytechnic Institute, Troy, NY 12180, USA; Bauers2@rpi.edu; 2Department of Chemical and Biological Engineering, Center for Biotechnology and Interdisciplinary Studies, Rensselaer Polytechnic Institute, Troy, NY 12180, USA; 3Departments of Biological Science, Biomedical Engineering, Center for Biotechnology and Interdisciplinary Studies, Rensselaer Polytechnic Institute, Troy, NY 12180, USA

**Keywords:** zoonotic diseases, infection, glycosaminoglycans, virus, COVID-19, SARS-CoV-2

## Abstract

Zoonotic diseases are infectious diseases that pass from animals to humans. These include diseases caused by viruses, bacteria, fungi, and parasites and can be transmitted through close contact or through an intermediate insect vector. Many of the world’s most problematic zoonotic diseases are viral diseases originating from animal spillovers. The Spanish influenza pandemic, Ebola outbreaks in Africa, and the current SARS-CoV-2 pandemic are thought to have started with humans interacting closely with infected animals. As the human population grows and encroaches on more and more natural habitats, these incidents will only increase in frequency. Because of this trend, new treatments and prevention strategies are being explored. Glycosaminoglycans (GAGs) are complex linear polysaccharides that are ubiquitously present on the surfaces of most human and animal cells. In many infectious diseases, the interactions between GAGs and zoonotic pathogens correspond to the first contact that results in the infection of host cells. In recent years, researchers have made progress in understanding the extraordinary roles of GAGs in the pathogenesis of zoonotic diseases, suggesting potential therapeutic avenues for using GAGs in the treatment of these diseases. This review examines the role of GAGs in the progression, prevention, and treatment of different zoonotic diseases caused by viruses.

## 1. Introduction

Zoonotic diseases are infectious diseases originating in animals, either symptomatic or asymptomatic, that can be passed to humans through exposure incidents referred to as spillovers [1]. Viruses transmitted through animals cause many of the world’s most problematic and contagious diseases [2]. HIV passes from apes to humans, Hendra passes through horses to humans, and Ebola passes from bats to humans. There have been eight major outbreaks globally that were caused by zoonotic spillovers since 1990 [3] (Table 1). As humans encroach further on natural habitats, these spillovers become more likely [4]. With the recent COVID-19 pandemic, the need for scientific research and understanding of zoonotic illnesses has become imperative. An overview of global outbreaks of zoonotic diseases during the last half-century is shown in Table 1.

GAGs are negatively charged, linear polysaccharides including heparin (HP)/heparan sulfate (HS), chondroitin sulfate (CS)/dermatan sulfate (DS), keratan sulfate (KS), and hyaluronic acid (HA) (Figure 1). With the exception of hyaluronic acid, GAGs are covalently linked to core proteins found anchored to animal cell membranes. HA is usually found linked to the cell surface through non-covalent binding to HA-binding proteins such as CD44 and RAHMM [11].

By interacting with various proteins, GAGs play critical roles in myriad pathological and physiological processes, such as embryonic development, inflammation, cancer, cardiovascular diseases, and infectious diseases. The GAGs, attached to a core protein, comprise proteoglycans found on the host cell surface can serve as co-receptors for pathogens, facilitating specific interactions between host and pathogen during the early stage of infection by bacteria, fungi, viruses, and parasites [12] (Table 2). Recent studies taking place during the COVID-19 pandemic show that the interactions between cellular HS and S-protein of SARS-CoV-2 are critical for viral infection [13,14]. Thus, manipulating GAG-protein interactions may lead to effective therapeutic approaches to defending host cells from infection. In this review, we focus on the role of GAGs in the pathogenesis of viral zoonotic diseases.

## 2. Zoonotic Diseases

### 2.1. Filovirus—Ebola

Ebola virus is a filovirus with a negative RNA structure [6]. There are six distinct species of the Ebola virus, but so far, only Zaire, Reston, Sudan, am Bundibugyo have caused outbreaks in humans (Table 1) [7]. Ebola is classified as a hemorrhagic fever. Marburg is an older, more well-known hemorrhagic fever in the same family as Ebola. Initially, Ebola presents with flu-like symptoms but rapidly progresses with dehydration, hallucinations, and uncontrollable bleeding [6,40,41]. This hemorrhaging is how most health care workers, as well as family members, are infected. Quarantine and proper disposal have led to relatively smaller outbreaks occurring every few years since the most common route for infection is person to person through bodily fluids and direct contact [5].

Ebola is a viral hemorrhagic fever originating in Africa. The earliest strain is known as Ebola Zaire. The first outbreak of Ebola Zaire occurred in 1976, known as Zaire (now the Democratic Republic of the Congo (DRC)), near the Ebola River. Ebola was 88% fatal to its victims [42]. Since the initial discovery of the Ebola virus and Ebola virus outbreak in 1976, six distinct strains have been identified, and there have been almost two dozen outbreaks across the globe [5]. Most of these outbreaks centered in South and Central Africa, with a few outbreaks appearing elsewhere (Table 1). Traditional burial rights were changed to avoid contagion spread, and all infected materials were buried or bleached. This allowed all the outbreaks in Central Africa to remain small and isolated [42]

West Africa, however, was not well prepared [42]. The quarantine methods that saved West Africa during smallpox outbreaks were insufficient for containing Ebola, a disease in which the patients are spewing infected material as they die [40]. In the rural parts of Africa, death rites are still practiced that require special care and burial ceremonies for the deceased. These instances in which people must touch the infected corpses led to the large increase in case numbers seen in the infamous West African Ebola outbreak in Sierra Leone from 2013 to 2016, resulting in over 30,000 cases with almost 12,000 fatalities [40,41]. This outbreak led to many new treatment options and preventative measures being tested for Ebola, including the first approved Ebola vaccine in late 2019, but five years later, there is still no effective treatment for this debilitating disease. The West Africa outbreak was the first time that sexual transmission played a major part in the spread of Ebola. Sexual transmission is now thought to be a contributor to one-half of global Ebola flare-ups [43]. A second large outbreak recently occurred from 2018 to 2020 in the DRC with over 3000 cases [36].

The large time gaps in between Ebola outbreaks suggest there is a reservoir host, most likely a mammal, which carries Ebola without becoming symptomatic. Ebola has been found in primates across Africa, but these animals usually expire quickly once infected, so that cannot be the reservoir host [44,45,46]. As seen in the 1989 Ebola Reston outbreak in the U.S., not all strains of Ebola are fatal in humans, but these are always fatal in nonhuman primates [7]. A typical reservoir animal should not become very ill when infected but instead remain capable of traveling large distances in Africa, causing outbreaks across the continent. One animal that has been implicated as a possible reservoir animal for Ebola is fruit bats. These bats are widely distributed throughout Africa and roost in caves where bodily fluids of one individual are easily transferred to others that leave these bodily fluids throughout Africa where primates and people can come into contact with them [46]. The Kitum cave in Kenya was found to be a source of Marburg in the 1980s when two tourists became ill. However, more recently, bats found in the area of Ebola outbreaks have been traced back to Kitum cave, where all the samples collected from the cave came back positive for Ebola [46].

### 2.2. Henipavirus—Hendra and Nipah

Hendra and Nipah are both paramyxoviruses (genus *Henipavirus*), but they are very different from the more well-known member of this family, measles [47]. Typical paramyxoviruses such as measles and mumps can be deadly in children but are very rarely serious in the general population. Hendra virus was first seen in 1994 in southern Australia [8]. To date, there have only been seven human cases of Hendra virus infection [48]; however, this disease does pose a serious threat to the horse racing industry [45]. Animals develop fevers, inflammation, bloody froth at the nose and mouth, seizures, and refusal to eat or drink and humans become infected only on close contact. One man died in 1994 after having close contact with several infected horses in his stable. An outbreak in New South Wales in 2019 led to a vaccine for horses being developed, but there are few therapeutic options for infected horses and humans [8].

Humans infected with Nipah show symptoms similar to those observed in Hendra, including inflammation, seizures, respiratory distress, and confusion [9]. Nipah is also a *Henipavirus* in the paramyxovirus family, which was first found to infect humans and pigs in Malaysia in 1998. Nipah is often fatal in humans but less so in pigs [45]. In the initial outbreak, which spanned from September 1998 to May 1999, there were almost 300 cases in Malaysia with a mortality rate of around 40% [9]. There have been several outbreaks of Nipah across Malaysia, Bangladesh, and India. Overall, the outbreaks in Bangladesh from 2001 to 2015 had around 80% mortality. The most recent in the southern part of India began in 2018 and has been 91% fatal [9]. It appears that as time has passed, Nipah has become more fatal. An outbreak in West Bengal was 100% fatal, but only five people were infected. Animals with Nipah antibodies have been found in Africa, although there have not been any outbreaks in Africa as yet [49]. Nipah is a serious concern to public health in certain regions showing spillover.

Hendra virus infection is most concerning due to its recent appearance in horses in Australia. Horses in recent outbreaks become quite ill, suggesting there may be another animal serving as a reservoir host. Gray-headed flying foxes are fruit bats that live in all of Australia but mostly in Queensland, where the outbreaks began [50]. Flying foxes all across Australia have tested positive for Hendra virus infection but do not become ill [51]. Across Australia, there are also humans who work with bats and are bitten, scratched, and have come into close contact with bat bodily fluids, but these humans do not become sick. Horses are an important intermediate host since only humans in close contact with Hendra-infected horses become ill. Horses can contract the virus from the bats directly, but horses apparently act as amplifiers for the virus, promoting human infection.

Similarly, bats in Asia seem to have always had the Nipah virus in them, but there was no outbreak until 1998. The amplifier host for Nipah appears to be pigs. These pigs can contract the virus by eating fruit and vegetation, which have been contaminated by bodily fluids from the infected bats. Unfortunately, due to limited treatment options in 1998, it was necessary to cull over 1 million pigs to halt the spread of Nipah across Asia. Annual outbreaks in Bangladesh, however, are not due to the pigs; in this case, people ingest fruit and palm sap from trees where infected bats are roosting [51]. Nipah may have required pigs only for the initial crossover, but they are no longer required, making Nipah even more dangerous.

### 2.3. Coronavirus—SARS, MERS, and COVID-19

Severe acute respiratory syndrome (SARS)-coronaviruses (CoV) are named for their halo shape when observed under a microscope (Figure 2). SARS initially spilled over in 2003 in southern China and caused global panic. The severe respiratory symptoms were fatal and very contagious. Over 8000 people were infected, of which over 700 died. The outbreak lasted about one year and then died out. Everyone infected either died or recovered by late 2004. SARS can be spread through the air in sneezes and coughs or through contact with bodily fluids. Once the virus was in humans, it took less than 24 h for the infection to spread across the ocean to North America. SARS presents in a similar manner as pneumonia and can cause respiratory failure.

Middle East respiratory syndrome (MERS) was first observed in 2012 in Saudi Arabia and Jordan. Like SARS, observed 9 years earlier, MERS is a coronavirus. MERS was much less lethal than SARS, and most people infected with MERS either showed no symptoms or minor cold-like symptoms. MERS-associated deaths were the result of rare cases in which MERS led to pneumonia or kidney failure [10]. Humans become infected with MERS in the same way that humans were infected with Hendra. Instead of caring for sick horses, MERS patients were caring for sick camels who spread the virus in their coughs and sneezes. To date, there have been less than 100 human deaths caused by MERS [53]. Humans usually contract MERS from infected camels, but since camels become ill when infected by the virus, they are probably not the reservoir for MERS.

COVID-19 is caused by severe acute respiratory syndrome-related coronavirus 2 (SARS-CoV-2). The first known case was reported in Wuhan, China, in December 2019. Since COVID-19 transmits easily human-to-human through air contaminated by droplets and small airborne particles, this disease has spread very fast, leading to an ongoing worldwide pandemic. As of October 2021, it has caused over 219 million confirmed COVID-19 cases and 4.55 million associated deaths worldwide. In the U.S., one out of 500 residents have died of COVID-19. Older adults and people who have severe underlying medical conditions such as heart or lung disease or diabetes are at higher risk for developing more serious complications from COVID-19 illness. Although we currently have excellent vaccines and a few therapeutics available to fight this disease, it is unknown how long the efficacy of the vaccines can last and how effective the current vaccines are for viral mutants.

The reservoir host for SAR-CoV, MERS, and SARS-CoV-2 are believed to be bats that are found in large numbers throughout the world [45]. Bats have a higher level of interferons present in bats as compared to other animals, including humans. These cytokines, generated by host immune cells, enhance the initial immune response toward infection, promoting rapid viral clearance and persistent infections [54,55,56]. Moreover, bat lungs are rich in HS GAG capable of coronavirus binding [57]. Thus, in zoonotic infection, bats are believed to be the initial host serving as a viral reservoir.

### 2.4. Lentivirus—AIDS

There are two well-known lentiviruses that infect humans, human immunodeficiency virus (HIV)-1 and HIV-2. While very similar, these are distinct viruses from distinct sources. HIV-2 was discovered after HIV-1 was discovered and was formerly known as simian immunodeficiency virus (SIV) when it was found in monkeys. The main symptoms of HIV-2 in humans appear to be a weakened immune system, diarrhea, and weight loss. HIV-2 cases thus far have all stemmed from people who are infected by close contact with *Cerocebus atys*, the sooty mangabey, a common gray-colored monkey that inhabits the forests of West Africa, particularly Senegal [45]. HIV-2 is both less infectious and far less lethal than the more virulent lentivirus than HIV-1.

HIV-1 is the virus responsible for acquired immunodeficiency syndrome (AIDS) and the decades-long pandemic [6]. The AIDS epidemic reached global awareness in the 1980s, but spillover is thought to have taken place sometime in the 1970s in Africa. Further research has suggested that the first case of HIV-1 in humans was actually in 1959, involving an English man known only as the Manchester sailor corresponding to the true patient zero [45]. HIV-1 and subsequently AIDS presents with severely lowered T-cell levels as well as overall immune weakness, leading to lethal infection with usually non-lethal bacteria and viruses. A common cold can kill an AIDS patient, and most AIDS patients will die of some type of pneumonia as their bodies cannot fight the infection.

Lentiviruses are not like paramyxoviruses or coronaviruses that can be spread on a sneeze. The exchange of bodily fluids between the infected patient and the non-infected person is required to contract HIV-1 or HIV-2. Blood or other infected fluid needs to enter the bloodstream of the non-infected person for the virus to infect [6]

HIV-1 and HIV-2 are both found naturally in the primates of West and Central Africa. HIV-2 is thought to have evolved from the similar but slightly different SIV, which is common and deadly in primates [58]. When mice carrying human CD4+ T cells and other immune cells are exposed to SIV and bred with or exposed to infected mice, a drastic drop in CD4+ T-cell levels is observed in the final generation of mice when compared to the initially infected generation [58]. This suggests that although SIV initially spilled over into humans, it then mutated into the HIV-2 we now see in humans and primates. In contrast, HIV-1 started out in chimpanzees in Central Africa, where it was asymptomatic until people prepared and ate bush meat containing the virus. Since then, HIV-1 has run rampant throughout the human population.

### 2.5. Flaviviruses—Dengue, Encephalitis from Ticks, Japanese Encephalitis, Zika

Flaviviruses are positive single-stranded RNA viruses that commonly pass from ticks and mosquitos to humans [59]. The Flaviviridae family is responsible for dengue fever, tickborne encephalitis, Japanese encephalitis, and Zika virus. All these flaviviruses use GAGs in their initial binding to host cells [60].

Dengue fever is one of the most virulent and lethal diseases carried by mosquitos, with over 20 million cases per year globally [60]. Mild cases of dengue result in high fevers, muscle aches, and fever-induced delirium. Severe cases can cause major hemorrhaging and death, leading to this virus’ other name, dengue hemorrhagic fever. Dengue fever passes to humans through a mosquito bite and is commonly found in tropical climates where there is a large mosquito population. Humans are currently the only known reservoir for dengue fever, and it is passed when a mosquito bites an infected individual and within a week bites a second healthy individual in its next blood meal. Although dengue originated in monkeys in Africa, it has been around for centuries, and there are records of human cases as far back as the 1700s. The first human case is believed to have resulted when a mosquito bit an infected monkey and then a human. Despite its origins, dengue almost exclusively infects humans now. Despite this, there is no effective treatment or vaccine for dengue [61]. In 2001 a study to analyze the virus population across the Brazilian amazon showed four different strains of dengue along with many other deadly viruses [62].

Tickborne encephalitis is a severe neurological disorder caused by a flavivirus. Although the initial symptoms are similar to those of the flu (fever, chills, nausea, and vomiting), in most patients, the virus spreads to the central nervous system (CNS) and causes long-lasting neurological damage [63]. With three subtypes (far Eastern, Siberian, and European), tickborne encephalitis can show 20% fatalities, with the remaining 80% of infected individuals having long-lasting CNS effects [63]. While there is a vaccine for the European strain, unfortunately, due to climate change and the subsequent uptick in tick population globally, cases are steadily increasing.

Japanese encephalitis is responsible for almost 50,000 deaths per year [64]. Symptoms of Japanese encephalitis mirror tickborne encephalitis with fever, delirium, and muscle pain. Just like tickborne encephalitis, when not fatal, Japanese encephalitis often results in lifelong neurological damage [65,66]. Japanese encephalitis played a role in the delayed response to the initial Nipah outbreak in Malaysia. When Nipah originally broke out in pig farmers in Malaysia, it was misdiagnosed as Japanese encephalitis. This is probably due to the similar initial symptoms and the fact that the pigs and bats in Asia are the natural reservoir hosts for Japanese encephalitis [45].

Zika virus was first discovered in humans in 1947, and although it causes very mild symptoms in adults such as fever and rash and in rare cases, infected adults can develop Guillain-Barre syndrome due to Zika infection. The most concerning aspect of Zika infection is the severe birth defects observed in the fetuses of infected pregnant women, and these birth defects prompted the World Health Organization (WHO) to declare Zika a global health concern in 2016 [24]. Zika was originally detected in monkeys in Uganda but has since become a global threat with cases throughout Africa, the Americas, and Asia. Zika, such as dengue, is spread through the bite of an infected mosquito.

### 2.6. Orthohepevirus—Hepatitis E

The first reported human cases of hepatitis E were in India in 1978 [67]. In 1997 hepatitis E was discovered in domestic pigs, confirming its origins as a zoonotic virus. Since then, rats have been discovered to be the original carrier of hepatitis E [68]. Hepatitis E can be asymptomatic but typically presents with abdominal pain, nausea, vomiting, and fever, eventually leading to liver failure. Hepatitis E is typically spread by ingesting contaminated drinking water or other poor sanitation situations [69]. Every year there are around 20 million individuals infected with hepatitis E globally [70]. Although rats may have the oldest strain of hepatitis E, the strain that pigs carry is much more likely to cause modern spillovers and subsequent human infections. Approximately 10% of the pigs in slaughterhouses across the U.S. carry hepatitis E [70]. In addition to pigs and rats, sika deer and nonhuman primates also carry hepatitis E [67].

### 2.7. Lyssavirus—Rabies

Rabies cases have been recorded since 2000 BCE, and this single-stranded negative RNA virus is observed globally, with approximately 60,000 human rabies deaths annually [71,72]. Most humans are infected with rabies through the bite of an infected animal, most commonly a dog, although other mammals, including bats and cats, can also transmit the virus to humans. Rabies symptoms include hydrophobia, aerophobia, staggering, confusion, agitation, fever, and foaming at the mouth. Sadly by the time these symptoms are displayed, the virus has progressed beyond the point of recovery [73]. Rabies vaccines, proper containment of infected animals as well as rapid human treatment options have significantly reduced the impact of rabies on society. Even so, rabies remains in the top 10 viral causes of human death [73].

## 3. Wet Markets and Bushmeat

Aside from people infected while caring for sick animals, many of these spillovers are due to a combination of bushmeat and wet markets. Bushmeat is the common name for meat that has been hunted or gathered from the forest/bushland in Africa and Australia [74]. Bushmeat is a staple in many African villages in environments that are not very friendly to farming livestock. Many times, villagers will eat dead bats and primates they either kill or find. This is thought to be how Ebola, HIV, and Lyssa virus initially crossed over. Almost every outbreak in Africa can be traced back to interactions (either eating or skinning) with dead primates and/or bats [46,75]. Consumption of an infected bat started the 2013 Ebola outbreak, which then continued through human-to-human spread. The DRC outbreaks generally stemmed from human consumption or contact with dead chimpanzees and gorillas [74]. The term wet markets is used to describe markets in which live animals are sold for food and either slaughtered on site or immediately after purchase [74]. Asian wet markets have been a source of global controversy for many years for selling live animals for human consumption and have been the cause of many outbreaks [76]. SARS is believed to be initially transferred from bats to another small mammal called a palm civet, which was then eaten. Following the 2003 outbreak, palm civets across China were culled in large numbers as they were thought to be the source of this virus. The involvement of the civet has been brought into question but not before the palm civet population was severely damaged [45]. Wet markets also commonly sell wildlife animals as food. This is often where the infection issue comes to light. These wild animals are taken out of their natural habitat and are kept in small, overcrowded cages with other animals and other species. Bodily fluids of many species mix into the food and water of the animals being sold. Wildlife markets are both an answer to food scarcity, food deserts, and the continuation of traditional recipes. Food deserts in Asia have significantly shrunken in recent decades. Despite this, almost 80% of consumers choose to buy from wet markets in order to obtain fresher meat or less common meat, such as from reptiles, small rodents, and bats [74,76]. Following the Ebola outbreak in 2013, the WHO came out with an R & D blueprint that provides a framework for immediate research and response during viral outbreaks [77]. This plan was used during the coronavirus outbreak in 2020. Because of the effectiveness of this plan, two vaccines were developed during the pandemic to prevent further deaths.

## 4. Roles of GAGs in Pathogenesis of Viral Zoonotic Diseases

### 4.1. Ebola

Filoviruses such as Ebola need to bind to glycoproteins on host cells to infect [17]. The viral glycoprotein (GP) has two subunits that aid in the infection of host cells. GP1 is responsible for the initial binding of the virion to the host cell, while GP2 is key in membrane fusion between the virion and the host cell [78]. Soluble GAGs such as heparin have shown inhibition of the infection of filoviruses in human cells [79]. Cells treated with heparin in any concentration showed competitive inhibition of viral infection [79]. Using timed addition studies, heparin was found to clearly inhibit GP1 and prevent initial binding to the host cell [78]. Further studies using heparin’s effect on viral infection and progression have shown that pretreated cells are significantly less likely to become infected with the Ebola virus. When cells are treated with HP solution, the subsequent exposure to the Ebola virus does not produce infection [80].

### 4.2. SARS, MERS, and COVID-19

SARS-CoV-2, like all coronaviruses, attaches to host cell protein receptors through its spike glycoprotein, and this binding is mediated binding to HS on the surface of the host cells [5,6]. SARS-CoV-2 surface is decorated with envelope (E), membrane (M), and spike (S) proteins. A virion lands on the host cell surface by binding to HS proteoglycan (PG). S-protein undergoes proteolytic digestion by host cell surface protease, which initiates viral-host cell membrane fusion by conformational change caused by host cell receptor binding (HSPG and ACE2). ACE2 is an established host cell surface receptor in SARS-CoV-2 host cell entry. Virion enters the host cell and further experiences proteolytic processing by endosomal host cell protease [57,81].

SARS-CoV-2 causes respiratory illnesses and shows significant binding to human lung cells, and the most common proteoglycan in the human lung is HS. Therefore treatment with HP results in competitive inhibition of SARS-CoV-2 infection of human lung cells [13]. Most of the studies that have centered on HP and HS and their relation to SARS-CoV-2 infection relied on surface plasmon resonance (SPR) to estimate the amount of HP needed for lung cells to escape infection. The percent inhibition of infection of lung cells is linearly related to the log of HP concentration, and a relatively small amount of heparin is required for 100% inhibition of infection [82]. Viral infections produce an inflammatory response in the body as the immune system attempts to fight off the infection. As seen in sepsis and other widespread infections, this inflammatory response can lead to increased blood coagulation [83]. HP is an anticoagulant and also shows anti-inflammatory and antiviral activity, suggesting that it may be an excellent candidate for the prophylaxis and treatment of SARS-CoV-2 infection [81,83]. In severe cases of SARS-CoV-2, there is a leakage of proteins out of epithelial cells that are normally prevented by heparinase. Heparin can also prevent this by inhibiting heparinase activity in infected individuals [81]. Heparin has also been shown to negate the effects of circulatory histones resulting from viral infections [81].

### 4.3. AIDS

HIV-1 and HIV-2 both bind to the gp120 protein receptors on the surface of human host cells [6]. GAGs offer possible antiviral effects for these infections. HP does not normally bind to host cells, and when radiolabeled HP is introduced to host cells, there is little to no retention of the HP. However, in the presence of HIV-1 significant amounts of radiolabeled HP are retained on the surface of host cells [15]. These results suggest that HP binds to the glycoproteins on the surface of the viral particles and results in competitive inhibition. Other GAGs, i.e., CS, DS, and GAG mimetics, i.e., dextran sulfate, fucoidan, also are also known to interfere with viral infection. However, only highly sulfated HP and its close relative, HS, effectively prevent the attachment of HIV-1 virions to host cells [15]. Similarly, polyanions inhibiting the gp120 binding in the gut during early SIV and HIV-2 infection largely prevent infection and lead to immune system damage [84].

### 4.4. Dengue, Encephalitis from Ticks, West Nile Virus, Zika

Dengue fever, tickborne encephalitis, Japanese encephalitis, West Nile virus, Zika, yellow fever are all caused by viruses are in the same family that are collectively known as flaviviruses. All these viruses are positive RNA-based viruses. They all bind to negatively charged HS and positively charged DC-SIGN receptors on host cells. Negatively charged HP molecules represent ideal competitors for these receptors blocking infection [85].

Like all flaviviruses, dengue virus uses GAGs on the cell surface for initial attachment. This has led to extensive research into the possibility of using GAGs to treat or prevent dengue virus infection. The envelope protein on flavivirus envelopes is the first point of contact between virus cells and host cells [86]. Highly sulfated HP, as well as HP with reduced sulfation, were tested to compare their ability to inhibit the binding of dengue virus to host cells, the highly sulfated HP successfully prevented dengue infection in the test cells [61]. Further studies of the dengue virus have shown that HP binds the envelope protein at residues K291 and K295, blocking viral entry in host cells [87]. Suramin (Figure 1) is a well-known HP mimetic that shows significant inhibition of viral infection [88]. Suramin at various concentrations also shows noncompetitive inhibition of the dengue NS3 helicase with an IC_50_ of 0.4 [89].

Infections by different flaviviruses are inhibited by GAGs with different structures of chain lengths. Japanese encephalitis can only be inhibited by highly sulfated HP polysaccharides, whereas Zika is best inhibited by HP oligosaccharides [86]. HP inhibits Zika virions binding to host cells. Both sulfated and unsulfated HP do this in a dose-dependent manner. Unlike HP, which only prevents infection in the early stages of virus binding, suramin inhibits every stage of viral infection. As soon as cells are treated with suramin, the current stage of infection, be it binding, fusion, or replication, is halted [85]. The concentration of HS on the host cells exposed to Japanese encephalitis has an impact on the ability of a virus to result in plagues [90,91]. When HP chains of different molecular weights were tested for anti-Zika virus activity, a critical chain length could result in effective viral inhibition [92]. Similarly, GAGs have been shown to inhibit tickborne encephalitis virus infection of cell cultures by up to five-fold [93].

### 4.5. Hepatitis E

Hepatitis E has three open reading frames. Open reading frame 2 has been found to control the binding of virion particles to liver cells. Open reading frame 2 preferentially binds to the HS GAG chains of cell surface proteoglycans. This binding led researchers to test the binding compatibility between open reading frame 2 on hepatitis E to HP. When measured against the control protein, there was significant binding to HP. Therefore, HP might represent a possible antiviral treatment for hepatitis E through its competitive inhibition of open reading frame 2 binding [20].

### 4.6. Rabies

The virus causing rabies is a negative single-strand RNA virus that uses proteins on the virion to bind to host cell receptors. As with many other negative RNA viruses, HS mediates virion host cell binding, and when neuronal cells are pretreated with HS, the virus is incapable of infecting these pretreated cells [22]. As with many other zoonotic viruses, the more highly sulfated HP has excellent antiviral properties through competitive inhibition of virus binding to the less sulfated HS on the surface of the host cell [22]. An HP concentration of between 20 and 40 μg/mL shows excellent inhibition of rabies virus as determined using enzyme-linked immunosorbent assay [94].

## 5. Potential Prophylactic and Therapeutic Applications of GAGs in the Treatments of Viral Zoonotic Diseases

GAGs are polysaccharides with anionic, disaccharide repeating units and are found primarily on the surface of host cells [95]. One GAG that has garnered significant interest in antiviral studies is HS. This sulfated polysaccharide inhibits viral binding to host cells as well as viral penetration and subsequent infection [95]. The initial penetration stage of virus infection controls the virulency as well as the pathogenicity. This initial stage is mediated by the interaction between surface glycoproteins on the virion and the host cell receptors. As previously discussed, viruses initially bind to host cells by attaching to GAGs. HS, and similar GAGs, also act as mediators for endocytosis in viral infection. Treating virus particles with GAGs has been found to inhibit the binding of the surface glycoproteins to host cells, effectively neutralizing the virus (Figure 3) [93]. As early as the 1960s, studies on HS have shown its potential as antiviral activity. Heparinized blood has also been shown to inhibit infection by GAG-binding pathogens [96].

There have been many studies since the 2013 Ebola outbreak concerning the possibility of using glycochemistsy to prevent or treat Ebola virus infection [10]. Among these studies, the consensus has been that the T-cell production by CD8+/CD4+ has a direct relationship with the progression of Ebola infection. When nonhuman primates were exposed to the Ebola virus glycoprotein and subsequently depleted of their CD8+/CD4+ T cells, they became unable to have an immune response and succumbed to illness [10]. Tipton and coworkers mapped the glycopeptide to understand the interaction between glycoproteins present on the Ebola virus surface and the host immune system. GP-1 and -4 were shown to readily react with samples from Ebola survivors to produce large amounts of interferon gamma, which is an important cytokine produced by antigen-specific CD8+/CD4+ cells to fight infection, meaning these peptides could provide possible vaccine components to help the host immune system better cope with Ebola virus exposure [10]. Cai and coworkers tested the possibility of using immunotoxins to prevent or reverse Ebola virus infection. These immunotoxins have been shown to bind to the GP-1, which, as stated above, inhibits the progression of Ebola virus infection in humanized cells as well as nonhuman primates [5]. One of the main symptoms of Ebola infection is the violent widespread hemorrhagic due to poor cell adhesion. This lack of adhesion is caused by glycosylation of the surface glycoprotein. In cells infected with Ebola, where the enzymes are needed for glycolysis, the cell adhesion is unaffected [97]. If this glycolysis could be interrupted, the most common cause of death in Ebola patients (hemorrhage) would be eliminated. Since the 2013 West Africa outbreak Ebola virus has been found in the semen of surviving male Ebola patients [43]. GAGs are inherently nontoxic, making them an ideal candidate for preventative drug discovery. If people in areas where Ebola survivors are living could take GAGs as a preventative measure, this might eliminate the possibility of sexually transmitted Ebola, reducing the chances of another outbreak as the virions would be unable to bind to host cells and cause infection.

Dengue fever is also a hemorrhagic fever characterized by vascular injury and leakage due to an excess of NS1 dengue glycoprotein. NS1 binds to host HS; therefore, exogenously administered HS can mediate dengue infection and possibly be used to treat other hemorrhagic fevers [96]. HP has been avoided because its anticoagulant activity can contribute to hemorrhage but the HP mimetic suramin is currently being used off-label to treat dengue fever [98].

At present, the only treatment options for Hendra and Nipah are supportive therapy to treat the symptom and broad-spectrum antivirals designed for RNA viruses [9]. Based on the above mentioned, these options are insufficient and have been ineffective for preventing the increased mortality of Hendra and Nipah. For paramyxoviruses to infect, they first bind to the host cell and fuse their lipid envelope through the attachment and fusion glycoproteins binding to ephrin receptors on the host cell. Once these are bound, the hydrophobic fusion peptide is inserted into the cell to allow genome release. Measles is a paramyxovirus for which a vaccine is given to most children during infancy. This vaccine uses antibodies specific to the fusion protein on the measles envelope to prevent attachment to the host membrane [49]. In 2019 researchers tested a similar antibody treatment option for both Nipah and Hendra virus in primate and rodent subjects [49]. Treatment with these prefusion anti-fusion protein antibodies inhibited viral infection, reducing post-infection lethality. Ferrets given antibodies specific to the fusion protein avoided infection or, when given several days post infection, were able to recover [99]. This similar result across species speaks of a high likelihood of possible human analogs.

HP is a well-known blood thinner that activates antithrombin to prevent blood clots in humans and animals [19]. As with many other virus particles, Hendra and Nipah bind to HS on the surface of host cells. When heparin was added before and after exposure to Hendra and Nipah virus, HP showed competitive inhibition of Hendra and Nipah virus both [19]. When HP binds to the HS receptors on host cells, there is no or limited ability for Hendra and Nipah to show trans infection between host cells [19]

Hendra and Nipah are in the same family and genus but are distinctly different viruses (Figure 4). Although their glycoproteins seem to react similarly when treated with antibodies specific to the fusion protein, it should be noted that these studies all use anti-Hendra fusion protein antibodies against Hendra and antibodies specific to the fusion protein against Nipah [100]. Researchers discovered that the G protein is more important to Nipah infection than its fusion protein, whereas the fusion protein in Hendra is the main component of membrane attachment [100]. Molecular docking studies have been used to find common peptides that may inhibit the binding of glycoproteins on the surface of the Nipah virus [101]. Naturally occurring human antibodies have also been found to inhibit Hendra when given to infected animals [102]; interestingly, some of these monoclonal antibodies that neutralize Hendra also inhibit Nipah infection, with less reliability [102].

Sulfated marine glycans have shown a wide range of medical applications, including antiviral activity [108]. Since the natural host cell receptor, HS is a sulfated GAG using sulfated marine glycans as an antiviral treatment should inhibit virus binding. Sulfated glycans from marine sources have been found to effectively inhibit adenovirus in humanized cells [109]. Since Ebola, Hendra, and Nipah all use glycoproteins on their cell surface to bind to host membranes, these marine sulfated glycans could be a natural antiviral treatment option for these diseases. Rabies virus is inhibited by carrageenan, a sulfated marine glycan found in red seaweed [110].

## 6. Conclusions and Future Perspective

Zoonotic spillovers have only become more frequent over time, with approximately one in the year globally. The most recent outbreak of SARS-CoV-2 sparked a global pandemic on the same scale as the Spanish influenza pandemic. The increased frequency is due to many factors. The changing habitat of infected animals is being changed by climate change as well as human expansion. Additionally, the increasing world population and the subsequently increased need for food animals are encroaching on wild animal habitats and numbers at unprecedented rates. The strain that humans place on ecosystems is being felt globally. When these strains are not well managed, spillovers occur [62].

Eliminating wet markets, limiting bushmeat consumption, and habitat preservation can only do so much. Humans will continue to come into contact with infected animals, and spillovers will continue to occur. In conjunction with conservation efforts, research is required to better understand zoonotic diseases as well as new options for prevention and treatment.

HP and other GAGs could provide these options. HP has shown great promise as a prophylactic antiviral treatment [111]. With HP’s antiviral, anti-inflammatory and competitive binding to host cells, HP could be the next big antiviral treatment or prophylactic. HP is not the only GAG that has shown impressive activity in relation to SARS-CoV-2 treatment, low molecular weight HP, enoxaparin, showed a lower mortality rate with similar benefits HP in the treatment of SARS-CoV-2 patients [112]. Coronaviruses, filoviruses, flaviviruses, lentiviruses, and Henipavirus cause some of the most debilitating human diseases. The data already collected show that HP, suramin, or similar compounds have the potential to mediate if not eliminate these viruses in the human population.

Glycochemistsy has progressed greatly in recent years, along with GAG research, and recently, glyconanoparticles have been tested for antiviral activity [113]. Perhaps by coating the traditional gold and silver core nanoparticles or origami DNA nanoparticles with GAGs, an improved antiviral approach might be possible. For example, AuNPs have been shown to prevent binding at the DC-SIGN receptor, while HP prevents the binding of viral spike proteins to the HS receptors on host cells. Thus, coating the AuNPs with HP might inhibit binding at both the DC-Sign receptor and the HS receptor, effectively preventing viral infection.

GAGs are sulfated polysaccharide chains, so it stands to reason that other polysaccharides, having similar structures and sulfation content, might also provide similar results. Sulfated polysaccharides such as fucoidan are readily available in seaweed globally and could provide another treatment option [108]. GAGs and their derivatives (some of which have low or no anticoagulant activity) could represent a class of antiviral drugs against a myriad of viruses.

## Figures and Tables

**Figure 1 diseases-09-00085-f001:**
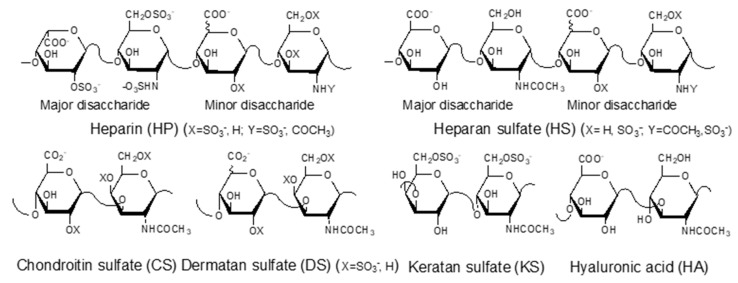
Structures of glycosaminoglycans (GAGs).

**Figure 2 diseases-09-00085-f002:**
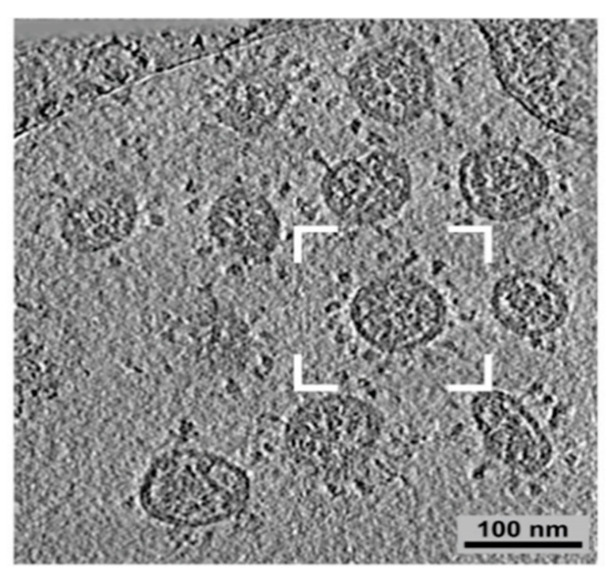
Coronavirus structure under microscope adapted from [52]. SARS-CoV-2 is another coronavirus of concern.

**Figure 3 diseases-09-00085-f003:**
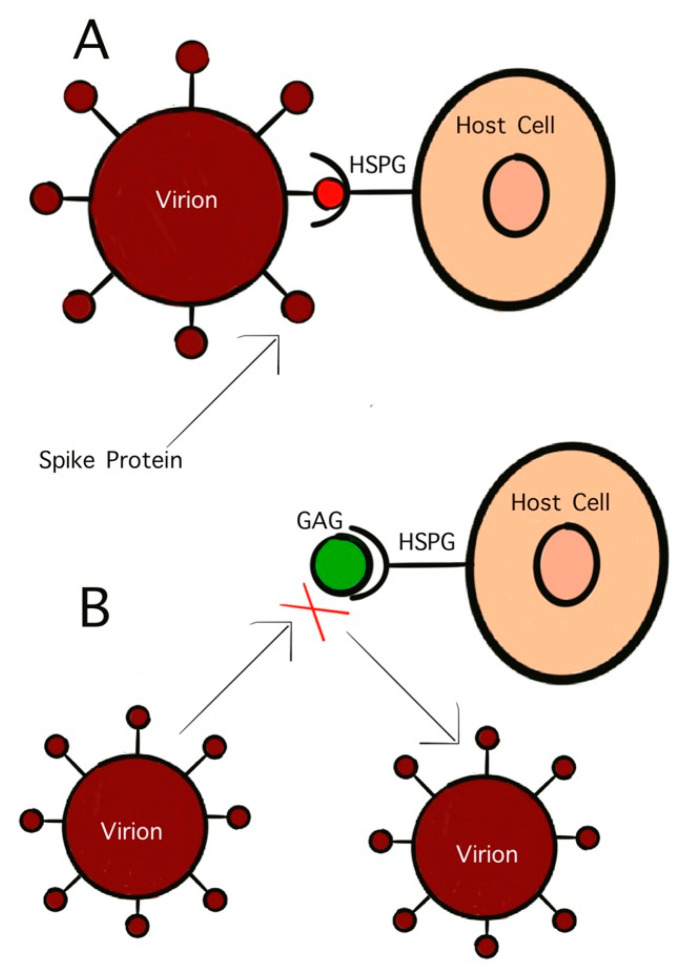
Competitive inhibition of virion binding to host cell receptor by GAGs. (**A**) Interaction of S-protein with HS on the host cell surface; (**B**) inhibition of HS mimetic on the interaction of S-protein with HS.

**Figure 4 diseases-09-00085-f004:**
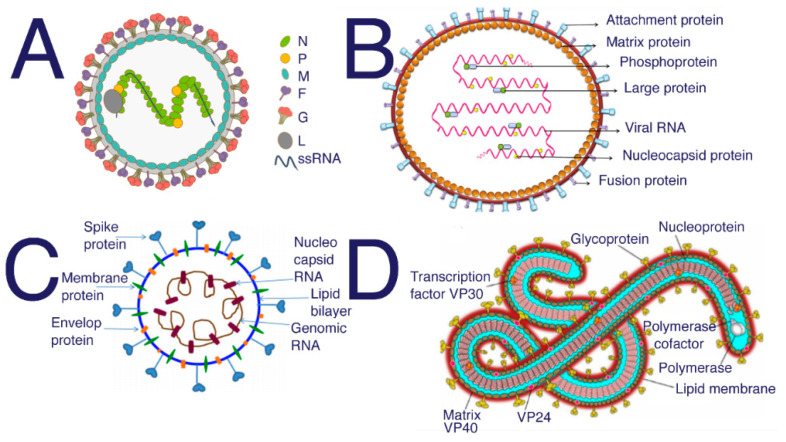
Structure of (**A**) Nipah from [103] (**B**) Hendra [104] (**C**) SARS-CoV-2 [105] (**D**) Ebola [106]. SARS, SARS-CoV-2, and MERS all require HS to act as a coreceptor in order to bind to host cells [5,6,48]. Many studies have been performed to investigate the inhibitory properties of HS against SARS and SARS-CoV-2. These studies have shown that sulfated polysaccharides exhibit similar inhibition [14]. Suramin is typically used as an antiparasitic treatment in humans; however, suramin shows significant inhibition of SAR-CoV-2 infection [107].

**Table 1 diseases-09-00085-t001:** Zoonotic diseases outbreaks by year, location, and strains [5,6,7,8,9,10].

Year	Location	Virus/Strain
1976	Zaire (Now DRC)	Ebola Zaire
1976	Sudan (Now South Sudan)	Ebola Sudan
1976	U.K.	Ebola Sudan
1989	Philippines	Ebola Reston
1990	U.S.	Ebola Reston
1994	Gabon	Ebola Zaire
1994	Mackay, Australia	Hendra
1994	Brisbane, Australia	Hendra
1995	DRC	Ebola Zaire
1996	South Africa	Ebola Zaire
1996	Russia	Ebola Zaire
1998	Malaysia	Nipah
1999	Cairns, Australia	Hendra
2000	Uganda	Ebola Zaire
2001	Gabon	Ebola Zaire
2001	Bangladesh	Nipah
2002	Republic of Congo	Ebola Zaire
2002	Bangladesh	Nipah
2003	Bangladesh	Nipah
2003	Guangdong, China	SARS
2004	Sudan (Now South Sudan)	Ebola Sudan
2004	Russia	Ebola Zaire
2004	Cairns, Australia	Hendra
2004	Townsville, Australia	Hendra
2004	Bangladesh	Nipah
2005	Bangladesh	Nipah
2006	Peachester, Australia	Hendra
2006	Muwillumbah, Australia	Hendra
2006	Bangladesh	Nipah
2007	DRC	Ebola Zaire
2007	Clifton Beach, Australia	Hendra
2007	Peachester, Australia	Hendra
2007	Bangladesh	Nipah
2008	Philippines	Ebola Reston
2008	Proserpine, Australia	Hendra
2008	Brisbane, Australia	Hendra
2008	Bangladesh	Nipah
2009	Bangladesh	Nipah
2010	Bangladesh	Nipah
2011	Uganda	Ebola Sudan
2011	Bangladesh	Nipah
2012	DRC	Ebola Sudan
2012	Bangladesh	Nipah
2012	Jordan	MERS
2012	Saudi Arabia	MERS
2013	West Africa	Ebola Zaire
2013	Bangladesh	Nipah
2014	Bangladesh	Nipah
2015	Bangladesh	Nipah
2015	Republic of Korea	MERS
2016	Bangladesh	Nipah
2017	Bangladesh	Nipah
2018	DRC	Ebola Bundibugyo
2018	India	Nipah
2019	Hunter Valley, Australia	Hendra
2019	Wuhan, China	SARS-Covid 19

**Table 2 diseases-09-00085-t002:** Common zoonotic viral, bacterial, fungal, and parasitic diseases.

Disease	Pathogen	Animal	Insect Vector	GAG-Binding Protein	Reference
**VIRAL**					
AIDS	HIV *Lentivirus*	Chimps	None	GP-120	[15]
Bird flu	*Influenza A H5N1*	Waterfowl	None	-	-
Chikungunya	*Alphavirus*				[16]
Covid/MERS/SARS	*Coronavirus*	Bats, civets/camels	None	Spike glycoprotein	[11,12]
Dengue fever	*Flavivirus*	Monkeys	Mosquito	Envelope protein	
Ebola	*Filovirus*	Bats	None	Filoviral glycoprotein	[17]
Encephalitis from ticks	*Flavivirus*		Tick	Envelope protein	[18]
Hendra	*Hendra Henipavirus*	Bats	None	Ephrin-B2 and -B3	[19]
Hepatitis E	*Orthohepevirus* HEV	Rats	None	ORF2 capsid protein	[20]
Japanese encephalitis	*Flavivirus*	Bats/pigs	Mosquitos		
Louping ill	*Flavivirus*	Sheep	Tick	-	-
Lymphocytic choriomeningitis	*Arenavirus*	Rodents	None	-	-
Mayaro	*Alphavirus Togaviridae*	Monkeys	Mosquitos		[21]
Nipah	*Nipah Henipavirus*	Bats	None	Ephrin-B2 and -B3	[19]
Orf infection	*Para poxvirus*	Sheep	None	-	-
Rabies	*Lyssavirus*	Bats		Attachment factor	[22]
Saint Louis encephalitis	*Flavivirus*				
Swine flu	*Virus, Influenza A H1N1*	Swine	None	-	-
Venezuelan equine encephalitis	*Alphavirus*				
West Nile virus	*Flavivirus*		Mosquitos	Envelope protein	[23]
Yellow fever	*Flavivirus*				
Zika	*Flavivirus*	Monkeys	Mosquitos	Envelope protein	[24]
**BACTERIAL**					
Anthrax	*Bacillus anthracis*	Hoofed animals	None	-	-
Bovine tuberculosis	*Mycobacterium bovis*	Cattle	None	Heparin-binding hemagglutinin	[25]
Brucellosis	*Brucella* sp.	Cows, goats, sheep	None	Unknown	[26]
Campylobacter infection	*Campylobacter* sp.		None	FliD protein	[27]
Cat scratch fever	*Bartonella henselae*	Cats	None	Pap31	[28]
Erysipeloid	*Erysipelothrix rhusiopathiae*	Fish, birds, mammals	None	-	-
Glanders	*Burkholderia mallei*		None	-	-
Leptospirosis	*Leptospira* sp.	Cattle	None	LigB adhesin	[29]
Listeria infection	*Listeria monocytogenes*	Ruminants, sheep	None	Surface protein ActA	[30]
Lyme disease	*Borrelia burgdorferi*	Deer	Tick	OspF-related proteins, adhesion BBK32, adhesion DbpA	[31]
Parrot fever	*Chlamydia psittaci*	Parrots and other birds	None	Unknown	[32]
Pasteurellosis	*Pasteurella multocida*	Domestic animals	None	OmpA β-barrel ion channel protein	[33]
Plague	*Yersinia pestis*	Rats and rodents	Flea	Ail outer membrane protein	[34]
Q fever	*Coxiella burnetii*	Sheep, goats, cattle	None	-	-
Rocky Mountain spotted fever	*Rickettsia rickettsii*	Rodents, dogs	Tick	Unknown	[32]
Tularemia	*Francisella tularensis*	Rodents, rabbits	Tick or deerfly		
Zoonotic diphtheria	*Corynebacterium diphtheria*	Dogs	None	-	-
**FUNGAL**					
Ringworm	*Tinea corporis*	Domestic animal species	None	-	-
**PARASITIC**					
Cryptosporidiosis	*Cryptosporidium* sp.	Calves and lambs	None	Mucin-like glycoprotein, CpClec	[35]
Giardiasis	*Giardia lamblia*	Domestic and wild mammals	None	Alpha-11 Giardin Annexin	[36]
Malaria	*Plasmodium falciparum*	Nonhuman primates	Mosquito	Circumsporozoite protein	[37]
Toxocariasis	*Toxocara canis* or *T. cati*	Dogs/Cats	None	-	-
Toxoplasmosis	*Toxoplasma gondii*	Cats	None	Protein of 104 kDa (P104) Microneme-2 (MIC2)	[38]
Trichinellosis	*Trichinella* sp.	Pigs	None	Unknown	[39]

## Data Availability

Not applicable.

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
