# Peer review of "Implications of Glycosaminoglycans on Viral Zoonotic Diseases"

_diseases, 2021, doi:10.3390/diseases9040085_

Round 1
Reviewer 1 Report
The review "Implications of Glycosaminoglycans on Viral Zoonotic Diseases " by Bauer, Zhang and Linhardt submitted to Diseases brings interesting points of view and revision of the current literature regarding a diverse range of viruses and their implications in human diseases. It is going to add to the current literature, and it will be of use not only for researchers but also, in my opinion, to health workers as well as policy makers. Nevertheless, the review misses relevant information regarding some of the described viruses that were performed in other regions of the globe, such as Latin America and the Caribbean. This is true for dengue, zika, yellow fever, among others, including animal models to study those diseases, such as congenital Zika syndrome in newborn squirrel monkeys. Also, they miss most relevant information of spill overs coming from animals to humans that are described for the Amazon region, published in well know journal, by well known scientists. I strongly believe that the review should mention and include references related to this topic to be accepted for publication. The authors should also mention the R&D Blueprint from WHO which is a global strategy and preparedness plan that allows the rapid activation of research and development activities during epidemics.
Some papers that can lead to the literature regarding Latin America and the Caribbean. 1) American Journal of Tropical Medicine and Hygiene 30:149-160, 1981; 2) American Journal of Tropical Medicine and Hygiene 30:674-681, 1981; 3) CRC Handbook Series in Zoonoses; Section B. Viral Zoonoses (G. W. Beran, ed.), pp. 214-217, 1994, Boca Ratón: CRC Press; 4) Virus Review and Research 4:48-49, 1999; 5) Cadernos de Saúde Pública, Rio de Janeiro, 17 (Suplemento):155-164, 2001
Author Response
Response: We thank reviewer #1 for his/her positive and constructive comments on our manuscript.
Specific points:
- 'SARS-CoV-2 pandemic all started with human interacting 16 closely with infected animals'. Ultimately, this hasn't been proved - maybe saying this is the dominant theory;
Response: This statement has been revised to clarify that this interaction has not been proven.
- References missing around SARS and SARS-CoV-2 in relation to GAGS;
Response: References have been updated in the revised manuscript.
- Figure 1 could be reduced - maybe focus on the relevant GAGs for the paper;
Response: Figure 1 has been reduced in our revised manuscript: we only show the structures of relevant GAGs.
4. Figure 3 needs to be remade - not very elegant crop&paste. It says it's been adapted though.
Response: We decided to remove Figure 3, instead, we present a short paragraph to state the potential applications of heparin in anti-COVID-19: “
“HP is an anticoagulant and also shows anti-inflammatory and antiviral activity suggesting that it may be an excellent candidate for the prophylaxis and treatment of SARS-CoV-2 infection”
5'SARS-CoV-2 surface is decorated with envelop (E), membrane (M), and spike (S) proteins. A virion lands on host cell surface by binding to HS proteoglycan (PG). S-protein undergoes proteolytic digestion by host cell surface protease, which initiates viral-host cell membrane fusion by conformational change caused by host cell receptor binding (HSPG and ACE2). ACE2 is an established host cell surface receptor in SARS-CoV-2 host cell entry. Virion enters the host cell and further experience proteolytic processing by endosomal host cell protease.' - this whole paragraph needs references.
Response: References have been updated.
Reviewer 2 Report
The manuscript by Bauer is indeed sound and brings a comprehensive review of how GAGs play a major role during viral infections. It brings a nice narrative around different viruses and highlights recent developments around cell surface GAGs as receptors for SARS-CoV-2 and how the use of heparin could prevent it. On the latter, it misses though the early work done by Mycroft-West (originally preprinted in early 2020).
Specific points:
- 'SARS-CoV-2 pandemic all started with human interacting 16 closely with infected animals'. Ultimately, this hasn't been proved - maybe saying this is the dominant theory;
- References missing around SARS and SARS-CoV-2 in relation to GAGS;
- Figure 1 could be reduced - maybe focus on the relevant GAGs for the paper;
- Figure 3 needs to be remade - not very elegant crop&paste. It says it's been adapted though.
- '
SARS-CoV-2 surface is decorated with envelop (E), membrane (M), and spike (S) proteins. A virion lands on host cell surface by binding to HS proteoglycan (PG). S-protein undergoes proteolytic digestion by host cell surface protease, which initiates viral-host cell membrane fusion by conformational change caused by host cell receptor binding (HSPG and ACE2). ACE2 is an established host cell surface receptor in SARS-CoV-2 host cell entry. Virion enters the host cell and further experience proteolytic processing by endosomal host cell protease.' - this whole paragraph needs references.
Author Response
- The review "Implications of Glycosaminoglycans on Viral Zoonotic Diseases " by Bauer, Zhang and Linhardt submitted to Diseases brings interesting points of view and revision of the current literature regarding a diverse range of viruses and their implications in human diseases. It is going to add to the current literature, and it will be of use not only for researchers but also, in my opinion, to health workers as well as policy makers. Nevertheless, the review misses relevant information regarding some of the described viruses that were performed in other regions of the globe, such as Latin America and the Caribbean. This is true for dengue, zika, yellow fever, among others, including animal models to study those diseases, such as congenital Zika syndrome in newborn squirrel monkeys. Also, they miss most relevant information of spill overs coming from animals to humans that are described for the Amazon region, published in well know journal, by well known scientists. I strongly believe that the review should mention and include references related to this topic to be accepted for publication. The authors should also mention the R&D Blueprint from WHO which is a global strategy and preparedness plan that allows the rapid activation of research and development activities during epidemics.
Response: We thank reviewer #2 for the positive and useful comments on our manuscript. Additional information from R&D Blueprint from WHO has been added to our revised MS.
- Some papers that can lead to the literature regarding Latin America and the Caribbean. 1) American Journal of Tropical Medicine and Hygiene 30:149-160, 1981; 2) American Journal of Tropical Medicine and Hygiene 30:674-681, 1981; 3) CRC Handbook Series in Zoonoses; Section B. Viral Zoonoses (G. W. Beran, ed.), pp. 214-217, 1994, Boca Ratón: CRC Press; 4) Virus Review and Research 4:48-49, 1999; 5) Cadernos de Saúde Pública, Rio de Janeiro, 17 (Suplemento):155-164, 2001
Response: Thanks for the list of literature. As you suggested, we fixed the miss information regarding Latin America and the Caribbean and cited the related papers.
Round 2
Reviewer 1 Report
The new version of the manuscript incorporates the suggestions.